# Role of Endogenous Glucocorticoids in Cancer in the Elderly

**DOI:** 10.3390/ijms19123774

**Published:** 2018-11-27

**Authors:** Emira Ayroldi, Lorenza Cannarile, Sabrina Adorisio, Domenico V. Delfino, Carlo Riccardi

**Affiliations:** Department of Medicine, Section of Pharmacology, University of Perugia, 06129 Perugia, Italy; lorenza.cannarile@unipg.it (L.C.); adorisiosabrina@libero.it (S.A.); domenico.delfino@unipg.it (D.V.D.); carlo.riccardi@unipg.it (C.R.)

**Keywords:** glucocorticoids, cancer, elderly, genomic instability, immunosenescence

## Abstract

Although not a disease itself, aging represents a risk factor for many aging-related illnesses, including cancer. Numerous causes underlie the increased incidence of malignancies in the elderly, for example, genomic instability and epigenetic alterations that occur at cellular level, which also involve the immune cells. The progressive decline of the immune system functions that occurs in aging defines immunosenescence, and includes both innate and adaptive immunity; the latter undergoes major alterations. Aging and chronic stress share the abnormal hypothalamic–pituitary–adrenal axis activation, where altered peripheral glucocorticoids (GC) levels and chronic stress have been associated with accelerated cellular aging, premature immunosenescence, and aging-related diseases. Consequently, changes in GC levels and sensitivity contribute to the signs of immunosenescence, namely fewer naïve T cells, poor immune response to new antigens, decreased cell-mediated immunity, and thymic involution. GC signaling alterations also involve epigenetic alterations in DNA methylation, with transcription modifications that may contribute to immunosenescence. Immune cell aging leads to decreased levels of immunosurveillance, thereby providing tumor cells one more route for immune system escape. Here, the contribution of GC secretion and signaling dysregulation to the increased incidence of tumorigenesis in the elderly is reviewed.

## 1. Introduction

Aging is a physiological process characterized by a progressive reduction of the body’s ability to maintain the homeostatic balance of cells, organs, and systems, and is associated with progressive alterations of the physiological systems [1]. The progressive functional decline of all organs that occurs during aging is a major risk factor for many diseases, such as osteoporosis, neurodegenerative syndromes, diabetes, arthritis, and cancer [2,3].

Both epigenetic and genetic factors interact with each other, contributing to a cumulative cascade process to induce aging phenotypes. Although the succession of biological events, and therefore the exact pathogenic mechanism responsible for aging, is still unknown, the identification of the so-called aging hallmarks may help to better define and study the molecular mechanism of aging and the aging-related diseases [1,4]. Historically, various theories have alternated to explain aging as caused by a single factor. It is now known that aging may be considered a cascade process, in which primary damage, or the primary hallmarks of aging (epigenetic dysregulation [5] and/or mitochondrial dysfunction, damage to the DNA [6] and/or the telomere [7]) induce secondary damage, i.e., the secondary hallmarks of aging, such as stem cell exhaustion (which can affect age-related diseases, such as sarcopenia, osteoarthritis, and osteoporosis), chronic inflammation, and nutrient signaling dysfunctions [8]. Overall, a general characteristic of aging is the imbalance between stressful stimuli and the ability of the organism to mount an effective response to the stressor stimuli and repair damage. At least two main features characterize this concept: Inflammaging, and the aging of the immune system, termed immunosenescence [9]. Both can affect cancer development [10]. Inflammaging is a state of sterile, chronic, low-grade inflammation characterized by increased activation of proinflammatory transcription factors, such as nuclear factor kappa B (NF-κB) and increased inflammatory cytokines, including tumor necrosis factor alpha (TNF-α), interleukin 1 (IL-1), and IL-6 [11]. Immunosenescence involves a remodeling of the immune system functions, leading to the decline of some functions, while others remain unchanged [9,12].

Cellular senescence is part of the aging process, but may be induced by other causes, e.g., stressful stimuli, such as ionizing radiation, chemicals, and oncogene activation. Cellular senescence is the irreversible proliferative arrest that is biologically necessary to ensure that DNA mutations are not perpetuated in the genome. It should be considered a tumor suppressor mechanism [13,14]. However, senescent cells, although in cell cycle arrest, are metabolically active and produce secretory proteins (senescence secretome), interacting with their microenvironment and acting on non-senescent cells to induce proliferation, constituting a link between cellular senescence, aging, chronic inflammation, and cancer [15,16].

One theory by which the progressive functional decline that characterizes aging has been partially explained is based on psycho-neuro-endocrino-immunology, which explains how some characteristics of aging are similar to those induced by chronic stress. According to this theory, dysregulation of the hypothalamic–pituitary–adrenal (HPA) axis, crucial for immune system homeostasis, may be responsible, at least in part, for the hallmarks of aging, including immunosenescence [17]. Immunosenescence, in turn, contributes to the increase of malignancies in the elderly [18].

This review highlights the current efforts toward understanding how glucocorticoid (GC) deregulation may affect tumorigenesis in the elderly.

## 2. Glucocorticoids in the Elderly

GCs are steroid hormones produced by the adrenal gland and regulated by the HPA axis. GCs have the significant physiological role of regulating critical homeostatic functions, including metabolism, immune system, cell growth, and development [19]. Although GC secretion follows a circadian rhythm, stressful stimuli can also regulate its production [20]. In fact, GCs are released as a result of stressful conditions, along with other stress factors, such as catecholamines and other neurotransmitters. GCs acting on the target organs and the metabolism constitute an important part of the body adaptive response to stress [21]. However, when stress becomes chronic, the adaptive, functionally useful response becomes harmful, and GCs may contribute to the immunophenotypic changes observed under stress conditions, which are similar to those seen in aging [22]. Aging is, in fact, characterized by a primary dysregulation of the HPA axis, which synergizes with the alteration of GC production due to chronic stress; the latter is often frequent in aging [23] (Figure 1).

Administered at pharmacological doses, exogenous GCs represent the mainstay of current anti-inflammatory and immunosuppressive strategies [24]. In oncology, GC use has been a hallmark of lymphoid cancer treatment given their ability to induce apoptosis, although not all patients are responsive to GCs [25]. In solid tumors, the role of GCs continues to be discussed and is often unknown [26]. GCs are included in some therapeutic protocols (e.g., prostate and breast cancer), and used as adjuvants to control the adverse effects of chemo/radiotherapy [27].

Most GC effects are genomic and mediated by glucocorticoid receptor (GR), a transcription factor that regulates gene expression either by binding to GC response elements in the promoters of the target genes or by interacting with other transcription factors, such as activator protein-1 (AP-1) or NF-κB, inhibiting their gene expression function [28,29].

GCs have a crucial physiological role in immune cell development, functions, and trafficking [30]. In the presence of increased levels of GCs, the immunomodulation of physiological doses of GCs becomes immunosuppressive. Consequently, the increased GC levels, due to age-related dysregulation, stress, or therapeutic chronic administration, together with other neuroendocrine factors [31], is crucial for driving and controlling the immunosenescence processes [22,23,32] (Figure 1).

### 2.1. Immunosenescence

Immunosenescence is the functional decline of immune cells that reduces their ability to respond to foreign and self-antigens [33]. The process involves natural (monocytes, natural killer cells, dendritic cells (DCs)) and acquired (B and T lymphocytes) immunity; involvement of the latter is greater. Hematopoietic stem cell (HSC) senescence, which reduces HSC self-renewal and HSC ability to commit to the various lineages, partly drives the mechanism of immune aging. In fact, both lymphoid organ involution and defective lymphoid precursor production are hallmarks of immunosenescence [34]. This leads to a significant change in the composition of T lymphocytes in peripheral lymphoid organs. Specifically, memory CD4 T cells were found increased, whereas γ/δ T cells and naive CD4 and CD8 T cells decreased. Humoral immunity is also functionally compromised in the elderly, since peripheral B cell number does not decline with age, but there is lower antibody response, decreased high-affinity antibodies, and decreased IgG isotype class switching [35].

There is a significant contradiction in natural immunity during aging. Namely, its regular functions at baseline levels and its inefficiency when specific roles are required, such that the impaired balance between relatively conserved natural immunity and altered adaptive response, with increased inflammatory cytokine production, contributes to the inflammaging phenotype, which, with respect to immunosenescence, is the other side of the same coin that is aging [9,16,36].

GCs play a crucial role in these modifications. The immune system alterations in the elderly are similar to those observed following chronic stress [37], where GCs are increased together with other neuroendocrine factors or GC therapeutic treatment [38], such as changes in cell-mediated and innate immunity and changes in cellular trafficking, suggesting that GCs may contribute to the induction of immunosenescence [39].

Neuroendocrine imbalance of the HPA axis appears to be involved in age-related immunological changes [17]. In fact, healthy elderly people present exaggerated HPA axis activation, exhibiting increased levels of endogenous cortisol, and decreased levels of dehydroepiandrosterone (DHEA) compared to young adults [40,41]. According to some researchers, the increased cortisol is due to the loss of the HPA axis feedback mechanism for age-related hippocampal neuronal loss. This has an impact on GRs number and function, since it has been reported that there is a reduction in the number of GRs consistent across different strains in the hippocampus and hypothalamic paraventricular nucleus, and a reduced binding capacity of GRs in the hippocampus and anterior pituitary, associated with a resistance of HPA axis feedback. This reduction was found to be the consequence of the age-related excess of glucocorticoids that causes a loss of hippocampal neurons, and as a consequence, the reduction of GR numbers. However, the age-related reduction of GR numbers is controversial, since it has been reported that the imbalance of HPA axis is due to augmented pituitary function with increased corticotropes, and GR expression in aged compared to younger rats. According to other researchers, the increased cortisol is associated with increased inflammatory cytokines, i.e., IL-1, IL-6, and TNF-α, characteristics of inflammaging that, in activating the HPA axis, increase cortisol [42]. The increased cortisol corresponds to decreased DHEA levels, with a consequent increase in the cortisol/DHEA ratio. According to the neuroendocrine hypothesis of aging, an increased cortisol/DHEA ratio results in augmented immunosuppressive GC activity and contributes to immunosenescence [43] (Figure 1). DHEA is physiologically responsible for anti-GC effects by both modulating protein kinase C (PKC) signaling and controlling the alternative splicing of GR mRNA, with increased expression of the GRβ isoform in inhibitory activity [44]. Consequently, its decrease in the elderly potentiates the effects of cortisol. For the direct effects of GCs on the immune system, the increased cortisol/DHEA ratio has been associated with increased risk of infection and chronic inflammatory diseases, and decreased immunosurveillance against tumors, with increased tumor frequency and invasiveness [39,45,46]. Notably, damage produced by stress throughout life and chronic stress that the elderly often experience may synergize with elderly HPA-axis dysfunction, altering the mechanisms and pathways implicated in physiological processes, such as cell proliferation, differentiation, and apoptosis [39,47]. GCs induce atrophy of the thymus and other lymphoid organs, and inhibit T cell activation by reducing the activity of Lck (LCK proto-oncogene, Src family tyrosine kinase) and Fyn (FYN proto-oncogene, Src family tyrosine kinase), protein tyrosine kinases activated upon TCR (T cell receptor) engagement. By inhibiting the ERK, JNK and p38 MAPKs (mitogen-activated protein kinases) pathways are activated during T cell triggering [48]. Moreover, higher cortisol levels have been associated with decreased naïve T cells and increased memory T cells [22,49]. It has been suggested that the GC-induced upregulation of CD95 (Fas) in naïve T cells may be responsible for increased apoptosis. Fas cell surface death receptor/Fas ligand (Fas/FasL) receptor–ligand pairing accounts for the apoptosis of activated lymphocytes, which limits the expansion of an immune response. Its hyperfunction in elderly naïve cells could explain the aging-related impairment of cell-mediated immunity to new antigens [50,51]. Moreover, GCs increase the numbers of peripheral regulatory T cells (Treg), a subpopulation of T cells responsible for switching off T cell activation to prevent autoimmune syndromes [52,53]. In addition, GCs induce a T helper 1 cell (Th1) to Th2 shift in cytokine production and increase serum proinflammatory markers, which contributes to the inflammaging phenotype [30,54] (Figure 1).

Furthermore, GCs exert a physiological immunosuppressive role on cells of innate immunity, affecting macrophage and dendritic cell (DC) functions. Specifically, GCs can directly act on macrophages to induce the M2-like phenotype [55] and weakening the antigen-presenting function, decreasing major histocompatibility complex (MHC) class II and costimulatory molecule expression, decreasing proinflammatory cytokines (e.g., TNF-α and IL-12), and inducing anti-inflammatory cytokines, such as IL-10 [56]. However, aging may affect the ability of the antigen-presenting cells, i.e., DCs and macrophages, to process the antigen, migrate, secrete cytokines, regulate costimulatory molecule expression, and respond to stimuli in their environment. The few studies on the role of GCs in natural immunity in aging have mainly been in vitro, measuring, for example, the effect of dexamethasone and DHEA on DC differentiation [57]. The results are not always concordant, although it may be suggested that GCs in the elderly have an inhibitory effect on natural immunity [22].

Despite the increased cortisol secretion [41], the cellular responsiveness to GCs is often diminished in the elderly [58,59]. For example, dexamethasone (a synthetic GC) of lymphocytes from elderly people has reduced in vitro sensitivity compared to that of young adults, suggesting the development of resistance to GCs. Typically, the physiological response to GCs depends on the cell and tissue type, individual differences, and the quantity of the GR isotype, and may be modulated by various factors, including chronic administration, stress, and age [60]. In patients with asthma, many mechanisms could account for GC insensitivity, for example, reduced affinity of GCs to the GR, fewer GRs, more GR isoforms acting as negative dominants, or altered expression of molecules involved in GR signaling (AP-1, NF-κB) [61]. By analogy, the same mechanisms could account for GC-resistance in the elderly. Nevertheless, the final outcome of the balance between blood GC increase and lower GC sensitivity of some targets in aging is immunosuppression.

Many age-related diseases are associated with immunosenescence, i.e., increased susceptibility to infectious, autoimmune, and chronic inflammatory diseases, and increased incidence of neoplastic disease.

### 2.2. Immunosenescence and Cancer

Although immunosurveillance against cancer has constantly been discussed, scientists have been debating for many years about the role of the immune system in cancer development and spread. Currently, it is commonly accepted that the immune system can protect against cancer by mounting a specific immune response, especially against highly immunogenic tumors. Besides the crucial therapeutic goals, introducing drugs that modulate the immunoregulatory molecules (checkpoints) at the immune effectors–tumor microenvironment interface makes an important argument in favor of a potentially effective anti-tumor adaptive immune response from which the tumor cannot escape [62]. Unfortunately, immunosurveillance against tumors may decline drastically with aging, contributing to the increased number of tumors that characterize the elderly [48].

As seen above, immune cells have reduced functional capacity in aging. The decreased naïve CD8+ and CD4+ cells and increased memory CD8+ cells (some researchers opine that the latter is due to persistent antigenic stimulation by cytomegalovirus infections) impede the response of the elderly immune system to cancer neoantigens [63]. In contrast, the immune system may respond to the so-called shared cancer antigen (i.e., embryonic antigen), to which immune cells have previously been exposed. Despite these immunosenescence markers, it has been suggested that young and elderly patients have comparable clinical responses to immune checkpoint–targeting drugs [18]. Moreover, the chronic inflammation that characterizes the elderly contributes to immunosenescence with the secretion of cytokines and the production of reactive oxygen species, constituting a further mechanism of carcinogenesis by inducing both immunosuppression and tumor growth [9,36,64].

On the other hand, tumors arising in the elderly appear to have learned to escape the immune system through various mechanisms, inducing immunosuppression and incapacitation of the immune cells. Tumors, at all ages, learn to enhance Fas/FasL system function and expression, and it appears that GCs are involved in this mechanism [65]. This mechanism may be amplified in the elderly, since an increase in Fas/FasL expression is a common feature of aging in different tissues and diseases [66,67], and particularly on aged leukocytes [68] with augmented lymphocyte apoptosis. An additional mechanism of tumor escape that may also be present in the elderly is the increased tumor cell production of immunosuppressive lymphokines, such as TGF-β, IL-10, IL-6, and TNF-α [65]. The increase of systemic and tumor microenvironment Th2 lymphocytes producing IL-4, IL-10, and IL-6, and the decrease in Th1 lymphocytes producing interferon gamma (IFN-γ) or IL-2—GCs are responsible for modulating Th lymphocytes. This has been observed in elderly patients with bladder or uterine cervical cancer, again suggesting a role for GCs in immunosenescence-driven cancer [16,22].

### 2.3. Glucocorticoids and Cancer

As explained earlier, immunosenescence concerns not only the elderly, but also individuals that experience chronic stress (chronic stress leads to premature immunosenescence), suggesting that chronic stress may favor the aging-related pathologies, including cancer [38]. However, research interest has only recently focused on the role chronic stress may play in cancer growth and spread. The current studies focus on both aspects of the role of stress on tumors: the direct response of the neoplastic cell to stress, and the antitumor response of immune cells under chronic stress conditions. While the correlation between stress and tumorigenesis is unclear and/or less studied, there is more information on stress and tumor growth/spread, which does involve stress hormones, including GCs [32,39]. Some information on the role of GCs on tumor growth in old age derives from investigation of the role of GCs in tumor development in subjects under chronic stress. In fact, as it is difficult to separate the direct effects of GCs from the other causes underlying the increased frequency of cancers in the elderly, there have been few studies for establishing the contribution of endogenous GCs in the increase in aging tumors. However, it could be assumed that the effect of GCs on cancer cells in aging may be similar to that observed during chronic stress or GC treatment, which regulate neoplastic cell growth and dissemination through at least three major mechanisms: (1) direct regulation of cancer cell survival molecules and pathways; (2) indirectly affecting systemic and tumor microenvironment immune cells; and (3) induction of epigenetic modification [25,39].

Above all, the direct activity of GCs on neoplastic cell biology has been studied in relation to their pharmacological use on lymphoid cancers, such as acute and chronic lymphocytic leukemias, Hodgkin’s and non-Hodgkin’s lymphomas, and multiple myeloma. In epithelial tumors, GCs are used to control the adverse effects of chemotherapy [26]. The direct effect of GCs on epithelial tumor cell survival is related to the tumor type. Many studies have reported that GCs increase the survival and progression of many epithelial tumors by inhibiting, for example, the p53 [69,70] tumor suppressor protein and/or by activating the p38 MAPK or Akt signaling pathways responsible for GC-induced tumor cell proliferation [71]. An even more important mechanism of GC-induced control of tumor growth involves immunosenescence. As already discussed, it may have a determinant role in the diminished immunosurveillance against tumors [22].

GCs are linked to aging and chronic stress through another aspect, i.e., the influence of GCs on epigenetic regulation, that is, the process of gene regulation driven by interactions between the genome and microenvironment. Epigenetic change involves numerous processes, including methylation and conformational changes in the chromatin and/or histones [39]. These processes contribute to the regulation of gene transcription. Epigenetic changes are associated with aging, and chronic stress and aging can act synergistically in inducing epigenetic aging [72]. Although the molecular mechanisms underlying this process are not all clear, GCs play a crucial role in epigenetic changes, since they induce stable DNA demethylation in hepatocytes [73] and drive changes in methylation during aging of dexamethasone-regulated genes, showing enriched association for aging-related diseases. Therefore, chronic stress may cause cumulative damage and early epigenetic aging through excess GCs. Aberrant GC signaling may therefore contribute to changes in DNA methylation, and subsequently in aberrant gene transcription associated with the aging-related pathologies, including cancer [74] (Figure 1).

## 3. Conclusions

Cancer is a multifactorial pathology with etiopathogenetic origins that are only partly known. Its multifactorial nature implies that each tumor type has both specific etiological causes and risk factors common to all tumors that contribute to cancer onset and development. This is why each neoplasm may become a unique disease [75]. Aging is a risk factor for the onset of cancer [76,77]. The increase in the incidence of cancer in the elderly has led to the hypothesis that the decline of psycho-neuro-endocrine-immunological homeostasis is a more distinctive and specific etiopathogenetic underlying cause of tumors in the elderly [18]. Dysregulated cortisol secretion is one of the most important events in determining alterations of the homeostatic equilibrium of aging, contributing to the induction of hallmarks of aging. In fact, modifications to cortisol secretion and the cortisol sensitivity of target tissues lead to both immunosenescence and epigenetic changes in DNA methylation/transcription [32]. Although the molecular mechanisms of immunosenescence (and GC-induced immunosenescence) are not completely known, a plausible hypothesis associates GC-induced immunosenescence with GC-induced epigenetic changes [39]. The increased cortisol, due to chronic aging and chronic stress, induces epigenetic alterations of DNA methylation, which in turn can induce immunosenescence. In fact, younger and older individuals have markedly different DNA methylation and gene expression of CD4+ and CD8+ T cells, where aging CD8+ T cells have an increased number of methylation changes [78,79]. Immunosenescence is crucial for the mechanism of immune escape by cancer and is at least partly responsible for the increased incidence of tumors in the elderly [14,18].

Many questions remain open, such as: at this point in our knowledge on cancer in the elderly, does the GC paradigm represent a philosophical and scientific crossroad? In other words, if the burden of cortisol dysregulation is crucial in increasing the number of cancers in old age—yet, this burden is not well established—then could many of the aging-related diseases, including cancer, be prevented or slowed by pharmacologically modulating the HPA axis dysfunction? Answering this question warrants further assessment of the burden of GCs in cancer in the elderly, distinguishing it from other causes underlying the increased incidence of tumors in aging. In our opinion, answering this question is the challenge of the future.

## Figures and Tables

**Figure 1 ijms-19-03774-f001:**
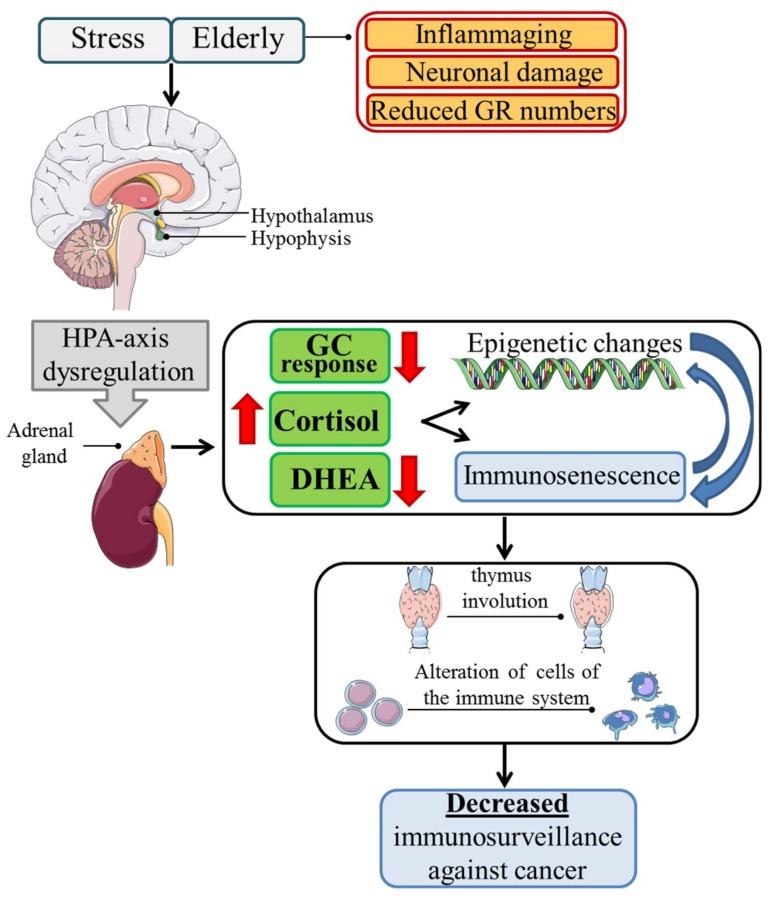
Aging and chronic stress share common mechanisms. Aging, characterized by hippocampal neuronal loss, glucocorticoid receptor (GR) reduction in the hippocampus and hypothalamic paraventricular nucleus, and inflammaging, together with chronic stress, induces hypothalamic–pituitary–adrenal (HPA) axis imbalance, with increased cortisol (red arrow, head up) and decreased dehydroepiandrosterone (DHEA) production and decreased response of target tissues to glucocorticoids (GCs). HPA-axis dysregulation contributes to both immunosenescence and epigenetic changes; the latter in turn aggravate immunosenescence (red arrows, head down), with the final result of thymic involution, immune cell alterations, and decreased immunosurveillance against cancer.

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
