# Peer review of "Role of Endogenous Glucocorticoids in Cancer in the Elderly"

_ijms, 2018, doi:10.3390/ijms19123774_

Round 1

Reviewer 1 Report

I have read with very interest your paper.

There are some inaccuracies in the writing of citations. Please correct ref 9 pag 6.

Please revise the references. The citations listed in references are written in different style

Author Response

I would like to thank the reviewer.  As suggested, reference 9 has been corrected as the reference's style.

Reviewer 2 Report

Altogether, the review is a bit superficial. There is a red line visible, and the arguments and the connections between different points are clearly presented. However, for some points it would be nice to elaborate a little bit more and to provide some details, especially for some of the proposed mechanisms.

Specific points:

1.       Line 28 “decline of all organs” – do you mean functional decline?

2.       In figure 1, “alterations of cells of the immune system” would be more clear. Moreover, it is not clear what about the GR reduction – only in the CNS or in the whole body? Also in the text, the point about GR reduction with increasing age remains vague. Are the underlying mechnisms known? Are there some hypothesis why the expression of the GR is reduced? When does this start?

3.       Line 81: “myelodysplastic disease”: what is the meaning of this phrase? Do you mean myelodysplastic syndrome (as outlined in line 232)? If yes, this is not necessarily related to cancer but rather describes impaired maturation of blood cells. May be one should focus on leukemia and lymphomas when speaking about GC-sensitive cancers.

4.       Line 108: “mature T cells”. Is this in contrast to immature thymocytes? It’s a bit puzzling, because in the next phrase it is said that the number of naive T cells (which are also mature) decreases. May be one could rephrase this sentence.

5.       Line 158-162: when describing effects of GC on APC, one should also mention that GC induce an M2 phenotype in macrophages. Thereis a whole bunch of literature about this phenomenon, and one macrophage phenotype is even named GC-induced M2.

6.       Line 174: “reduced affinity of the GCs to the GR”: it is not clear how this should work. Are there structural changes in the expressed molecules with age that the ligands are bound with a different affinity? If yes, this point should definitively better explained with the underlying molecular mechanisms.

7.       Line 204: increased FasL expression in tumors of elderly patients. The resons for this should also better explained, and references should be added for this phenomenon.

8.       The role of GC on epigenetic phenomena should be explained in more detail, including the molecular mechanisms.

9.       In the abstract, alterations in GC signaling are mentioned. There is no clear chapter about such phenomena, beside the vague argument of reduced GR levels (see point 2). Thus, for the reader this point is not quite clear.

Author Response

We thank the reviewer for the useful comments.  We changed the manuscript according to his/her suggestions in order to improve the paper and we hope now the reviewer may be sufficiently satisfied.

Specific points:

1.       Line 28.  Yes, we meant "functional decline".  Now this has been clarified in the text by adding the word “functional” (line 25).

2.       In figure 1 “alterations of immune system cells” has been replaced with “alteration of cells of the immune system”, as suggested.

The GR reduction was refered to the CNS system.  Now, this has been clarified: we added the words “in the hippocampus and hypothalamic paraventricular nucleus” in the figure’s legend and in lines 122-123.

As suggested, we have tried to address in a less vague manner the point regarding the GR reduction by adding the following sentences “This has an impact on GRs number and function, since it has been reported that there is a reduction in the number of GRs consistent across different strains in the hippocampus and hypothalamic paraventricular nucleus and a reduced binding capacity of GRs in the hippocampus and anterior pituitary associated with a resistance of HPA axis feedback.  This reduction was found to be the consequence of the age-related excess of glucocorticoids that causes a loss of hippocampal neurons and, as a consequence, a reduction of GR numbers.  However, the age-related reduction of GR numbers is controversial since it has been reported that the imbalance of HPA axis is due to augmented pituitary function with increased corticotropes and GR expression in aged compared to young rats” (lines 121-128).

3.       We agree with the reviewer that the use of “myelodysplastic disease” or “syndrome” was a mistake.  Thus, as suggested, we focused now on leukemia and lymphoma when we speak about GC-sensitive cancers.  In the text the sentence “has been an hallmark of lymphoid cancer treatment…although not all patients are responsive to GCs” replaced “is limited to myelodysplastic diseases,” (lines 80-81).  Additionally, the sentence “lymphoid cancers such as acute and chronic lymphocytic leukaemias, Hodgkin's and non-Hodgkin's lymphomas, multiple myeloma replaced the sentence “As GCs can induce apoptosis, they are used for treating myelodysplastic syndromes” (lines 223-224).

4.       As suggested, we rephrased the sentence of line 108: “This leads to a significant change in the composition of T lymphocytes in peripheral lymphoid organs.  Specifically, memory CD4 T cells were found increased, whereas γ T cells and naïve CD4 and CD8 T cells decreased…functionally compromised in the elderly, since peripheral B cell number does not decline with age, but there is lower antibody response, decreased high-affinity antibodies and decreased IgG isotype class switching” (lines 100-104).

5.       Line 158-162: As suggested, we now mention that GC induce an M2 phenotype in macrophages” by adding “Specifically GCs can directly act on macrophages to induce the M2-like phenotype and weakening the” with the relative reference (lines 153-154).

6.       Line 174: We meant that same mechanisms of GC-resistance developed in other medical conditions may develop also in the elderly.  To clarify this we added the following sentence “In patients with asthma, many mechanisms could account for GC insensitivity, for example, reduced affinity of GCs to the GR, fewer GRs, more GR isoforms acting as negative dominants, or altered expression of molecules involved in GR signaling (AP-1, NF-κB). By analogy, same mechanisms could account for GC-resistance in the elderly-“  (lines 167-170).

7.       Line 204: We added the following sentences and the relative reference to clarify on the role of FasL expression: “Tumor at all ages learn to enhance Fas/FasL system function and expression, and it appears that GCs are involved in this mechanism.  This mechanism may be amplified in the elderly since an increase in Fas/FasL  expression is a common feature of aging in different tissues and diseases and particularly on aged leukocytes with augmented lymphocyte apoptosis.  An additional mechanism of tumor escape that may be present also in the elderly isthe increased tumor cell production of immunosuppressive lymphokines such as TGF-β, IL-10, IL-6, and TNF-α” (Lines 196-201).

8.       The role of GC on epigenetic phenomena was now explained in more details “since they induce stable DNA demethylation in hepatocytes and drive changes in methylation during aging of dexamethasone-regulated genes showing enriched association for aging-related diseases” (lines 237-239).

9.       Since there is no clear chapter on the alterations in GC signaling, this mention has been eliminated in the abstract (line 13).